# Innovative Polymeric Biomaterials for Intraocular Lenses in Cataract Surgery

**DOI:** 10.3390/jfb15120391

**Published:** 2024-12-23

**Authors:** Kevin Y. Wu, Rebecca Khammar, Hafsah Sheikh, Michael Marchand

**Affiliations:** 1Department of Surgery, Division of Ophthalmology, University of Sherbrooke, Sherbrooke, QC J1G 2E8, Canada; 2Faculty of Medicine, Université de Montréal, Montréal, QC H3T 1J4, Canada; 3Faculty of Medicine, Queens University, Kingston, ON K7M 1G2, Canada

**Keywords:** intraocular lenses, polymeric biomaterials, nanomedicines, hydrogels, biosensors and biodetection, antibacterial materials, biocompatibility, shape memory polymers, surface modifications, cataract surgery

## Abstract

Intraocular lenses (IOLs) play a pivotal role in restoring vision following cataract surgery. The evolution of polymeric biomaterials has been central to addressing challenges such as biocompatibility, optical clarity, mechanical stability, and resistance to opacification. This review explores essential requirements for IOL biomaterials, emphasizing their ability to mitigate complications like posterior capsule opacification (PCO) and dysphotopsias while maintaining long-term durability and visual quality. Traditional polymeric materials, including polymethyl methacrylate (PMMA), silicone, and acrylic polymers, are critically analyzed alongside cutting-edge innovations such as hydrogels, shape memory polymers, and light-adjustable lenses (LALs). Advances in polymer engineering have enabled these materials to achieve enhanced flexibility, transparency, and biocompatibility, driving their adoption in modern IOL design. Functionalization strategies, including surface modifications and drug-eluting designs, highlight advancements in preventing inflammation, infection, and other complications. The incorporation of UV-blocking and blue-light-filtering agents is also examined for their potential in reducing retinal damage. Furthermore, emerging technologies like nanotechnology and smart polymer-based biomaterials offer promising avenues for personalized, biocompatible IOLs with enhanced performance. Clinical outcomes, including visual acuity, contrast sensitivity, and patient satisfaction, are evaluated to provide an understanding of the current advancements and limitations in IOL development. We also discuss the current challenges and future directions, underscoring the need for cost-effective, innovative polymer-based solutions to optimize surgical outcomes and improve patients’ quality of life.

## 1. Introduction

Cataract surgery is one of the most commonly performed surgical procedures worldwide, with intraocular lenses (IOLs) serving as a cornerstone in restoring vision by replacing the natural lens [1]. Over the years, advancements in biomaterials have revolutionized IOL design, with polymeric materials playing a pivotal role in meeting the stringent demands for durability, biocompatibility, optical clarity, and mechanical stability. The ideal IOL biomaterial must not only maintain high optical performance over the patient’s lifetime but also mitigate complications such as posterior capsule opacification (PCO), postoperative inflammation, postoperative infection (i.e., endophthalmitis), dysphotopsias, among others [1,2,3].

Recent innovations in polymer science have opened new avenues for optimizing IOL functionality. Developments in hydrogels, shape memory polymers, and light-adjustable lenses (LALs) exemplify how cutting-edge biopolymers are enhancing patient outcomes. These materials have enabled the integration of novel features such as sustained drug delivery, tailored optical properties, and superior resistance to PCO [4,5,6]. Despite significant advancements, challenges remain in achieving the perfect balance between optical performance, long-term stability, and cost-effectiveness.

Although numerous primary studies have investigated specific aspects of IOL biomaterials, a comprehensive, up-to-date review addressing the state of the art in polymeric innovations for IOLs is lacking. This review fills that gap by synthesizing recent findings on traditional and advanced polymeric materials, functionalization strategies, and emerging technologies. By evaluating clinical outcomes and patient satisfaction, this review aims to provide a holistic understanding of current trends and future directions in polymer-based IOL development. The findings underscore the potential of innovative polymeric biomaterials to redefine standards in cataract surgery and improve the quality of life of patients undergoing cataract surgery.

## 2. Essential Requirements for IOL Biomaterials

### 2.1. Biocompatibility

Biocompatibility refers to a material’s ability to interact with the surrounding tissue without causing any adverse effects. This means that the material should not induce blood clotting, tissue necrosis, anaphylaxis, foreign body rejection, or inflammation while still having all the desired properties such as flexibility and mechanical strength [7].

Biocompatibility is of paramount importance in IOLs as after they are implanted, they remain in the eye for decades and must continue to perform. IOLs are primarily in contact with uveal and lens tissue, which means that they cause a pathophysiological reaction in the uvea and lens epithelial cells (LECs). Thus, when discussing IOL biocompatibility, both uveal and capsular biocompatibility should be considered. Uveal biocompatibility is determined by the severity of postoperative inflammation in the eye [8].

During surgery, there is a breakdown of the blood–aqueous barrier (BAB). Then, the IOL causes an inflow of proteins and cells to the anterior chamber. Proteins that deposit on the surface of the IOL then facilitate cell deposition, which eventually results in a foreign body reaction against the IOL. The chemical structure and surface properties of the IOL determine the extent of this protein accumulation [3]. Uveal biocompatibility can be evaluated by examining the inflammation in the anterior chamber aqueous humor, clinically observed as cells and flare, and cellular deposits on the IOL surface [9]. Uveal biocompatibility is clinically relevant in the selection of IOLs for patients, particularly those with a compromised BAB, namely patients with a history of uveitis, pseudoexfoliation syndrome (PEX), or diabetes [10].

Capsular biocompatibility is determined by the interaction between the IOL and the remaining LECs in the capsular bag. The most common postoperative complication of cataract surgery, known as posterior capsule opacification (PCO), is caused by the remaining LECs proliferating, migrating, and differentiating into fibroblast-like cells on the surface of the IOL [9,11].

The biocompatibility of intraocular lenses (IOLs) is influenced by factors such as lens material and surface properties, optic edge design, and the haptic–optic configuration. Modifying these features can enhance both uveal and capsular biocompatibility, thereby reducing the risk of postoperative complications, including inflammation and posterior capsule opacification (PCO) [3]. 

### 2.2. Optical Clarity: Maintaining High-Quality Vision

Optical clarity is a measure of the spectrum an IOL transmits to the retina [12]. Increased optical clarity allows IOLs to reach higher levels of visual acuity. PCO, anterior capsule opacification (ACO), and glistening formation are all factors that may compromise the optical clarity of IOLs [13]. If the direct contact between the IOL and the remnant lens epithelial cells (LECs) leads to adherence, it may cause ACO and/or PCO. Considering this, the choice between square (or sharp)-edged or round-edged lenses is influenced by each lens’ ability to block migrating LECs and mitigate PCO (Figure 1) [2]. Maedel et al. collated 10 RCTs and concluded that sharp-edged IOLs are likely to be associated with less PCO formation than round-edged ones [2]. How this impacts optical clarity remains to be explored.

In order to manufacture IOLs with minimal fibrosis occurrence, the way the IOL may alter the stress field human lens capsule must be considered [14]. The IOL’s stress field describes the post-surgical mechanical stress distributed on or around an implanted IOL [14]. The presence of this stress field can affect lens position and stability [14]. This can influence visual outcomes and PCO risk [14]. Berggren et al. created the first 3D finite element model of a post-surgical human lens capsule with the IOL [14]. They used this to show that the changes in stress field distribution and magnitude can encourage LECs to elicit a fibrotic response, resulting in capsular changes postoperatively. This is especially relevant for accommodating IOLs [14]. Berggren et al.’s findings suggest that an IOL’s quality is influenced by its mechanical features, like sharp edges [2,14].

Concerns regarding optical clarity, and the biocompatibility of the biomaterials used, may arise upon the manifestation of signs like dysphotopsias. Dysphotopsias are unwanted visual phenomena that occur after cataract surgery, and are understood to be caused by either a glare from the IOL’s edge, or a dark crescent shadow [15]. Reduced clarity increases the risk of visual disturbances under low-light conditions, leading to patient dissatisfaction [15]. Due to their higher refractive index (RI), hydrophobic IOLs, especially acrylic ones, are suggested to be likely to be correlated to positive dysphotopsia (PD) versus hydrophilic IOLs [15]. This is because they have increased light reflection. However, Radmall et al. could not find a conclusive connection between IOLs’ RI and PD onset [16].

### 2.3. Mechanical Stability: Durability and Resistance to Deformation

For an IOL to be positioned correctly and remain in the intraocular environment for a long time, it must have mechanical stability. This includes resisting postoperative capsular bag contraction during healing and fibrosis [17]. IOLs must be flexible enough to fold and pass through a small incision. However, they also must be durable enough to unfold into the intended shape without deforming. They should not unfold too fast after insertion into the eye, for this may injure the capsular bag [13]. Mechanically unstable lenses risk being damaged during this process, leading to improper fit or function once inside the eye. However, a balance must be made between having increased IOL rigidity and achieving a smaller incision size—a rigid IOL would require a larger incision.

### 2.4. Resistance to Opacification: Preventing Secondary Cataract (PCO) Formation

IOL opacification can lead to glare, a reduction in contrast sensitivity, and severe visual impairment. Opacification typically appears within a few months after cataract surgery and can result in patient complaints and require IOL exchange [12]. PCO is the most common complication of cataract surgery, occurring in up to half of patients postoperatively. It can take weeks to years to arise [18]. There are two categories of PCO: the fibrotic form and the regenerative form.

Since surgery is still a controlled trauma to the eye, it induces a self-wound-healing response, which entails LEC proliferation. The residual LECs then begin proliferating in the anterior capsular regions, then migrating to the remaining IOL surface, including the posterior capsule. Ultimately, they encroach upon, and alter, the visual axis. The alteration of the visual axis is associated with fibrosis. Fibrosis is also interlinked with posterior capsule contraction and wrinkling, and matrix deposition. This represents the fibrotic form (Figure 2) [19].

The regenerative form of PCO occurs later on and is due to Elschnig’s pearls, swollen globular cells, that cause visual disruption by intruding on the visual axis [20,21]. The only effective treatment for PCO is neodymium-doped yttrium aluminum garnet (Nd:YAG) laser capsulotomy. Ursell et al. examined UK data on Nd:YAG incidence in 20,763 acrylic IOL-implanted eyes post-surgery, and estimated that 5.8–19.3% of patients will require Nd:YAG capsulotomy 5 years after cataract surgery. This estimate is influenced by factors like IOL type, patient age, and ocular comorbidity [22]. Although Nd:YAG capsulotomy is a common and relatively safe procedure, complications such as retinal detachment, elevated intraocular pressure, and endophthalmitis occasionally occur. The cost of the procedure itself, as well as the follow-up visits and management of potential complications, represent a considerable financial burden on healthcare systems [23]. It is therefore in our best interest to prevent PCO.

PCO rates vary with different IOL materials and designs. IOL haptics and optics design have been modified in attempts to prevent PCO. Sharp-edged IOLs have been associated with less PCO and lower Nd:YAG capsulotomy rates than round-edged ones. Having a sharp posterior optic edge seems to be an even more important factor in PCO prevention than the IOL material [2]. The Linnola sandwich theory states that bioactive materials prevent cell proliferation and capsular bag opacification because they allow a single layer of LECs to bond with both the posterior capsule and the IOL [24]. PMMA and hydrophilic acrylic IOLs have been associated with increased PCO severity, PCO rates, and Nd:YAG capsulotomy rates than hydrophobic acrylic IOLs. Hydrophobic acrylic IOLs also have lower PCO rates than silicone IOLs [25]. This is likely because hydrophobic IOLs adhere to collagen membranes, which leads to a tighter IOL apposition to the posterior capsular bag and stronger adhesiveness than other materials [26,27]. This shows the importance of the careful selection of IOLs to reduce complications and multiple interventions in patients, as well as the costs related to these treatments.

## 3. Types of Biomaterials Used in IOL Fabrication

### 3.1. Traditional Materials

Polymethyl methacrylate

Polymethyl methacrylate (PMMA) was the material used in the first ever IOL implanted in a human by Sir Harold Ridley in 1949. It is rigid (non-foldable), hydrophobic, and has a refractive index of 1.49. Its usual optic diameter is 5–7 mm.

PMMA IOLs have high uveal biocompatibility, a low inflammatory cell accumulation rate, and high visual quality [3]. However, they are intolerant to high temperatures and pressures. They are also associated with high PCO rates. One of the main disadvantages of PMMA IOLs is that they require a large incision for insertion. These larger incisions are associated with increased healing times and astigmatic refractive errors [28]. The development of phacoemulsification has allowed a decrease in incision size and favored smaller and more flexible IOLs [28]. Consequently, the use of PMMA IOLs has decreased significantly. Currently, they are mainly being used in low-to-middle-income countries due to their low cost [28,29].

Silicone

Silicones are a class of polymers mostly composed of covalently bonded silicon and oxygen atoms (siloxanes). One of the widely used silicones is polydimethylsiloxane (PDMS), which is hydrophobic, has a refractive index between 1.41 and 1.47, and requires an incision of at least 3.2 mm [30].

Silicone IOLs are foldable, highly transparent, heat-resistant, autoclavable, and moldable [30]. They are prone to both PCO and ACO, and calcification has also been noted in eyes with asteroid hyalosis following Nd:YAG capsulotomy (Figure 2) [28,30]. Silicone IOLs also have an increased risk of postoperative infections as they favor bacterial adhesion [30]. When they encounter the intravitreal air, they easily adhere to silicone oil droplets which negatively affects their transparency. This is an issue for patients with retinal detachment or diabetic retinopathy treated with silicone oil tamponade. Thus, silicone IOLs should be avoided in patients with a high risk of retinal detachment [31].

Silicone IOLs also pose an increased likelihood of injury to the capsular bag due to how quickly they unfold after insertion [31]. Therefore, silicone IOLs are not widely used due to the aforementioned complications negatively impacting patients’ visual acuity.

Acrylics

Acrylic IOLs have rapidly become the most popular choice of lens. They are foldable, which makes them ideal for microsurgery, and, in comparison to their PMMA or silicone counterparts, they have a significantly lower degree of PCO occurrence. They also unfold slower than silicone IOLs, which reduces the risk of capsular bag injury [30]. Acrylic IOLs are further divided into hydrophobic and hydrophilic on account of their water content. Hydrophobic IOLs have a water content ranging from 0.5 to 1%, while hydrophilic IOLs’ water content ranges from 18 to 38% [5].

Hydrophobic acrylic IOLs are very commonly used today [5]. They have an optic diameter of 5.5–7 mm and a refractive index of 1.44–1.55. They are available in one-piece or three-piece designs, and are most commonly made of poly(2-phenylethylmethacrilate) (poly PEMA), poly ethylmethacrylate (poly EMA), and poly(2,2,2-trifluoroethyl methacrylate) (poly TFEMA) [30,31]. Hydrophobic acrylic IOLs have lower rates of PCO than hydrophilic acrylic IOLs. This is because they adhere tightly to the capsule, thereby preventing LEC proliferation [32]. They also have significantly lower PCO rates than PMMA IOLs, but slightly higher rates than silicone IOLs; however, this does not seem to be clinically significant [33]. The main caveat of hydrophobic acrylic IOLs is the occurrence of glistenings. Glistenings are fluid-filled microvacuoles that can deteriorate a lens’ optical performance [34]. Nevertheless, a large number of glistenings are required to impact central image quality [35].

Hydrophilic acrylic IOLs are composed of hydroxyethyl methacrylate (HEMA), the same material used in contact lenses. They are flexible with a refractive index of 1.43. Their hydrophilicity allows them to be dehydrated and only require small surgical incisions below 2 mm. Hence, they are ideal for microsurgery [28,31]. Hydrophilic acrylic IOLs have a higher uveal biocompatibility versus hydrophobic acrylic IOLs, most likely owing to their high water content [3]. They are also less prone to postoperative inflammation [36]. However, hydrophilic acrylic IOLs have higher rates and increased PCO severity compared to hydrophobic acrylic IOLs [37]. Calcification is another problem that occurs with hydrophilic acrylic IOLs, and can impair vision [38].

### 3.2. Innovative Polymers in IOL Design

Hydrogels

Hydrogels have long been utilized in applications like contact lenses, corneal and scleral repair, and vitreous substitutes. This is due to their increased biocompatibility with the eye’s components. Often, these hydrogels have been used in conjunction with hyaluronic acid (HA). HA is a natural linear polysaccharide in the vitreous body. Manufactured HA-containing hydrogels have been shown to have increased physiological compatibility. Recently [39], hydrogels have emerged as cutting-edge polymers in IOL design, largely owing to their remarkable properties and adaptability. They are made of hydrophilic polymer chain networks that are capable of absorbing and retaining significant amounts of water. They have great biocompatibility, transparency, and flexibility, along with a soft-tissue-like consistency. They are remarkably similar to the human eye’s natural lens, and can mimic its physical properties [4]. An example of this is polyacrylamide–sodium acrylate hydrogels (PAHs), which have a refractive index of 1.41, and a tensile modulus similar to the human eye’s natural lens [40]. Interestingly, PAHs also enable continuous focal adjustment by the ciliary muscles [40]. Now, hydrogels can even be 3D-printed to customize the lens design and better address patient needs with personalized solutions [41].

Advancements in developing IOLs have involved optimizing hydrogels’ effectiveness. This entails incorporating hydrogels into acrylic IOLs. The outcomes have suggested that hydrogel-based hydrophobic IOLs have a lower PCO incidence than their hydrophilic counterparts [42]. Also, there has been some evidence suggesting that HA coating on hydrogel IOLs may improve their biocompatibility by lowering PCO rates [39].

Furthermore, applications have been proposed for hydrogel-based IOLs in the development of drug delivery systems to improve postoperative outcomes as they have a higher bioavailability than traditional ophthalmic drops. However, this would require careful patient selection and more extensive safety studies [43].

Ongoing hydrogel research has yielded promising advances that could introduce a more personalized era of customized IOLs, thus improving post-surgical outcomes.

#### 3.2.1. Shape Memory IOL

Shape memory polymers (SMPs) are polymers that irreversibly change shape when exposed to certain stimuli such as light, temperature, or pH. Hence, shape memory IOLs require smaller surgical incisions as they can be folded into a smaller size. They then revert to memory upon encountering changes in environmental conditions after insertion into the eye. These smaller incisions encourage shorter recovery times [1].

These polymers have a high uveal biocompatibility, very minimal or no glistenings, and high transparency. The cross-linker weight percentages and ratios of each monomer constituting the co-polymer can be adjusted to ensure that the SMPs have the desired mechanical and optical properties. Also, these numerical adjustments can ensure that the refractive index is similar to that of the human eye’s natural lens [44].

Song et al. created a solid SMP composed of ethylene glycol phenyl ether acrylate (EGPEA), ethylene glycol phenyl ether methacrylate (EGPEMA), and long alkyl chain. It was heated and shrunk in vitro, then expanded to a full-size IOL after reaching body temperature and remained stable. The refractive index was reported to be 1.514–1.499 and the SMP IOL exhibited good biocompatibility [45].

#### 3.2.2. Light-Adjustable Lenses (LALs)

Ametropia, or refractive error, is a very common complaint in patients post-surgery. Other ocular comorbidities, low preoperative visual acuity, and previous eye surgery were identified as potential risk factors for poor refractive outcomes after cataract surgery [46].

It seems likely that the incidence of postoperative refractive error will increase in the near future, since the patients who underwent corneal refractive surgery in the 1980s and 1990s have begun reaching the age where they will need cataract surgery. Due to these patients’ altered corneal structure, calculating their IOL power is more difficult and less accurate [47].

Previously, to adjust a postoperative refractive error, patients would have to undergo another invasive procedure such as corneal refractive surgery. However, light adjustable lenses (LALs) have changed this.

Alongside the innovative, cutting-edge polymers described above (hydrogels and shape memory IOLs), LALs have also made their mark (Table 1), largely due to their unique adjustability feature. LALs are the first IOL that can be adjusted post-insertion to fine-tune the patient’s vision. Their optics contain ultraviolet (UV)-sensitive silicone, and the haptic moiety is made of polymethylmethacrylate. The lens is inserted into the capsular bag via a small 2.8 mm incision and is irradiated to induce photopolymerization. If necessary, refractive changes of up to 3 D can be made this way. Once the desired refractive index is reached, a lock-in is performed where a final irradiation is used to permanently fix the lens’ refractive properties [30].

LALs have been found to have good biocompatibility and good visual acuity, stable refraction, and no IOL-associated pathologies or significant opacification. The UV exposure needed for LALs does not seem to result in any macular changes or predispose patients to macular edema or other macular pathologies because macular thickness measurements do not significantly change when taken preoperatively and postoperatively [48]. UV exposure during adjustment and the lock-in will likely be even less of a concern with the next generation of LALs, making them a very promising advancement in the field of IOL materials [49].

## 4. Functionalization Strategies to Enhance IOL Performance

### Surface Modifications

Potential surface modifications to optimize IOLs have been the topic of much recent research (Table 2). Hydrophobic IOLs adhere more tightly to the capsular bag than hydrophilic IOLs, thus preventing LEC migration and PCO occurrence [26,27].

Huang et al. showed that surface-grafting RGD (arginine–glutamine–aspartic acid) peptide, a cell adhesion molecule, onto hydrophilic acrylic IOL material, improves LEC adhesion. This strong adhesion prevents further LEC migration and division. Ultimately, this inhibits PCO, while not having a negative effect on the IOL’s optical and mechanical properties. This can be used to compensate for hydrophilic acrylic IOLs’ higher PCO rate compared to hydrophobic ones [50].

Wang et al. employed another strategy to prevent cell adhesion to decrease PCO incidence. They generated a hydrophilic surface coating of poly (sulfobetaine methacrylate) (PSBMA) on an IOL. The PSBMA brush surface-modified IOL seemed to effectively decrease PCO incidence [51]. Hydrophilic polyethylene glycol (PEG) was also immobilized onto a hydrophobic IOL via plasma-aided chemical grafting by Xu et al. This greatly inhibited the initial adhesion of LECs and prevented PCO, while not influencing the IOL’s optical properties [52].

Photodynamic therapy is an antiproliferative method to kill cells in which a photosensitizer (PS) is activated by light energy, and produces reactive oxygen species (ROS) that then inhibit cell proliferation. In a study by Tang et al., Chlorin e6 grafted α-cyclodextrin (α-CD-Ce6) was synthesized and assembled onto a poly (poly (ethylene glycol) methacrylate) (PPEGMA) brush-established IOL surface. This did not alter the optical properties or biocompatibility of the IOL. When the PS was exposed to light, it produced ROS and induced apoptosis. There was excellent PCO inhibition, safety, and biocompatibility in vivo [53].

Polydopamine (PDA) has been used as a photothermal modification. The near-infrared (NIR) light irradiation creates thermal energy and the elevated temperature of the IOL kills the LECs. These photothermal IOLs show efficient PCO prevention as well as good biocompatibility and optical properties [54].

Soaking the IOL in a drug can only carry a limited amount and is too quickly metabolized to meet the long-term demands of PCO prevention. To remedy this, drugs are combined with chemical materials on the IOL surface. Doxorubicin (DOX), an antiproliferative drug, has been loaded on a polydopamine (PDA) coating formed on the IOL surface. Hydrophilic 2-methacryloyloxyethyl phosphorylcholine (MPC) can then be grafted onto the IOL. MPC is water-soluble and composed of a phosphocholine group (hydrophilic subunit) and a methacrylate group (polymerizable subunit). The phosphate groups in MPC contribute to its biocompatible character. Also, though its structure allows it to trap water molecules and hygroscopic properties, its molecular complexity makes it difficult to form strong bonds with proteins and cells. Thus, it can inhibit bacterial biofilm formation, while providing the eye with ample lubrication [55]. The PDA (DOX)-MPC multifunctional coating released drugs slowly, lasting over 3 weeks, and was hydrophilic. This thoroughly and effectively prevented PCO by inducing cell apoptosis, and also demonstrated good safety and biocompatibility in the surrounding tissues [56].

Another antiproliferative drug, paclitaxel (Pac), might also inhibit PCO occurrence. A layer-by-layer (LBL) technique was used for the Pac-sustained release of hyaluronic acid (HA) and chitosan (CHI) multilayer-modified IOLs that demonstrated good, sustained drug release. This effectively inhibited LEC proliferation, therefore reducing PCO incidence [57].

Antimetabolic drugs have also been used in PCO prevention. Supercritical impregnation has been used to add methotrexate (MTX) onto the IOL surface. An 80-day drug release was achieved and an in vitro model showed the inhibition of epithelial–mesenchymal transformation, and thus might prevent PCO [58].

Immunosuppressant drugs such as cyclosporin A (CsA), protected by poly (lactide-co-glycolic acid) (PLGA), can be made by an economical spin coating technique. They showed a significant inhibition of cell proliferation, induced cell death, and prevented PCO in vivo [59].

## 5. Incorporation of UV-Blocking and Blue-Light-Filtering Agents

In vitro and in vivo studies have shown that UV light and blue light exposure may result in retinal photochemical damage, including damage to the retinal pigment epithelium and photoreceptors’ outer segments [60]. In light of this, considerations have been made to incorporate UV-blocking and blue-light-filtering elements in IOLs since the late 1980s [60]. In the human eye, UV light is absorbed by different structures based on its wavelength. UV light below 300 nm is absorbed by the vitreous and aqueous humors and the cornea, while UV light between 300 and 400 nm is absorbed by the natural adult lens [60]. Thus, IOLs are expected to achieve this level of filtration, and block radiation below 400 nm [61].

In vitro and in vivo studies have suggested that short-wavelength visible light, like blue light, can induce retinal phototoxicity. Thus, blue-light-filtering IOLs (BFIOLs) have been proposed [6]. Nevertheless, the overarching evidence in favor of using such IOLs, in light of any potential adverse effects, has been inconclusive, per two meta-analysis studies [6,62]. In fact, color vision was found to be compromised by blue-light filtering IOLs [62].

Kernt et al. conducted an in vitro experiment involving human retinal pigment epithelium to determine whether UV-blocking IOLs (UVIOL) and BFIOLs conferred protection against senescence, retinal damage, and oxidative stress [63]. The results suggested that both IOLs may provide such protection [63].

Clinical studies comparing UVIOLs and BFIOLs have indicated similarities between the IOLs’ visual acuity and color performance [64]. Also, patients with UVIOL implants are recommended to wear blue-filtering sunglasses, highlighting the need for combined UVIOLs and BFIOLs [64]. This also includes ensuring that such IOLs are biocompatible.

## 6. Integration of Drug-Eluting Properties

Postoperative endophthalmitis is a rare but very serious complication of cataract surgery that can lead to blindness [65]. It is mainly caused by Gram-positive bacteria that enter through the eye via surgical instruments or the IOL during surgery [66]. Currently, antibiotic eye drops are largely used to prevent endophthalmitis [67]. However, drug-eluting IOLs have the potential to prevent many postoperative complications such as inflammation and infection.

The antibiotic moxifloxacin (MXF) has been extensively studied in terms of drug-eluting IOLs. Its sustained release from the IOL is possible when an acrylic IOL is soaked in moxifloxacin for 4 days at 60 °C because this allows reversible endothermic interactions between the IOL and moxifloxacin. The concentration of MXF was found to be above the minimum inhibitory concentration (MIC) for *S. aureus* and *S. epidermidis* for at least two weeks. This is a relatively simple strategy that could potentially be applied and would reduce costs or the use of healthcare systems [68]. MFX was also shown to be non-cytotoxic against corneal endothelial cells [69].

This indicates that drug-eluting IOLs would be efficient in preventing acute postoperative endophthalmitis [70]. Supercritical solvent impregnation is another method used to make polymer-based drug delivery systems [71].

Norfloxacin and HEMA IOLs, cefuroxime and PMMA IOLs, and gatifloxacin in hydrophobic acrylic IOLs have been investigated for the prevention of endophthalmitis using this method [72,73,74]. The amounts of norfloxacin loaded and released were found to be sufficient to prevent the growth of common pathogens responsible for endophthalmitis [72].

Drug-eluting IOLs have also been studied for the prevention of inflammation. Dexamethasone (DXM) has been extensively studied for this purpose. Silicone IOLs soaked in DXM for 30 min then implanted showed a decrease in inflammatory markers compared to the control. However, maintaining sterility while soaking and drying the lenses was difficult [75].

## 7. Strategies to Mitigate Postoperative Inflammation and Infection

To remain in the eyes for long periods, it is essential that IOLs imitate natural lens’ features, including physical features, mechanics, biocompatibility, and long-term stability. Besides PCO and ACO, IOL implantation may also cause postoperative inflammation and infection [76]. Postoperative inflammation does not usually occur because of the IOL itself, since it is placed inside the capsular bag. Inflammation may occur as an immune response to cortex or nucleus pieces left in the eye post-surgery, or due to trauma that causes the IOL bag to rupture, exposing the ocular tissues to lens proteins. The surrounding inflammatory cells then adhere to the IOL and mount an immune response. As different types of cells adhere to the lens and form a biofilm, microorganisms can colonize the IOL surface and cause infection [74,77]. Since even biocompatible IOLs can cause some inflammatory cell adhesion, PCO often results postoperatively [78]. Treatment can be a deterrent to many patients due to the costly Nd:YAG laser capsulotomy [76]. Thus, it is better to develop IOL products with drug-eluting features.

Additionally, antimicrobial polymers can be added to IOLs [78]. Drug-eluting features may include releasing corticosteroids or NSAIDs, antibiotics, and antineoplastic agents into the posterior chamber [76]. Drug-eluting IOLs release drugs for weeks post-surgery, resulting in a longer-lasting effect [79]. This circumvents the need for patient compliance with eye drops. It also increases drug bioavailability in comparison to eye drops by delivering directly to the aqueous humor, thereby overcoming the corneal permeability barrier [79]. Drug-eluting IOLs can be created by being coated with polymers containing the active ingredient, or they can be soaked in the drug to take it up into the IOL matrix [76]. Drug release can be controlled chemically [76]. Refining this process could help mitigate postoperative inflammation and infection.

Qin et al.’s in vivo and in vitro study created an injectable, in situ curable, and adjustable IOL with a thermosensitive Poloxamer-based hybrid hydrogel [80]. It was successfully incorporated with antibacterial anti-inflammatory elements, suggesting potential for optimizing drug-eluting IOLs while mimicking the natural lens closely [78].

Drug-eluting IOLs have been shown to be successful in preventing PCO onset [81]. Yet a caveat is their time-consuming preparation, which involves extensive surface layering [81]. Chen et al.’s in vitro and in vivo study proposed using thermoresponsive agarose coating and doxorubicin hydrochloride (Dox) drug (which inhibits cell growth) to prevent PCO incidence by inhibiting LEC growth [81,82]. Agarose coating introduces a cell adhesion repelling characteristic to IOL coating surfaces, inhibiting LEC and PCO development [82]. The authors’ model relied on agarose’s water soluble nature at high temperatures (80–90 °C), and its ability to gelate at lower temperatures [82]. This enabled the authors to benefit from agarose’s (Aga) quicker preparation time, non-toxicity, low immunogenicity, high biocompatibility, and its sponge-like feature to store drugs [82]. In vivo assessments on New Zealand White rabbits with Dox@Aga coating showed no posterior capsule shrinkage or emergence of fibrosis in the eyes versus Aga-coated IOLs [82]. Also, Dox@Aga-coated IOLs manifested repelling of LEC adherence at posterior capsule [82]. This showed much promise for agarose-Dox-containing IOLs for optimized, timely and biocompatible IOL preparation and prevention of PCO onset [82].

Another approach was investigated by Lu et al., who utilized centrifugally concentric ring-patterned drug-loaded polymeric coating to prevent PCO [59]. This permitted the creation of biocompatible IOLs with steady drug release. The authors aimed to create an in situ drug delivery device [59]. They studied the outcomes of ring-patterned drug-loaded poly(lactide-co-glycolic acid) (PLGA) coating used to optimize spin coating parameters and drug-releasing properties [59]. Furthermore, some IOLs were also coated with cyclosporin A (CsA)-loaded coating (CsA@PLGA) to prevent PCO [59]. The cultured LECs suggested that the CsA@PLGA coating inhibited cell proliferation, indicating the potential translation of this model to real-life applications in creating PCO-resistant IOLs [59].

## 8. Clinical Applications and Outcomes

### 8.1. Visual Acuity

IOLs are not as elastic as natural lenses, meaning their implantation may lead to a postoperative loss of control over accommodation that achieves visual acuity [83]. Thus, recent research has focused on improving visual acuity, so accommodative and multifocal IOLs were introduced [83]. Accommodating IOLs allow for ciliary body contraction during near viewing [83]. Multifocal IOLs possess two or more focal points [83]. However, in their early post-surgical period, multifocal IOLs have decreased visual acuity and contrast sensitivity, often due to glare, halos, and light dispersion [83,84].

Kim et al. conducted a 12-month evaluation of the impact of using WIOL-CF, an accommodating sharp-edged IOL which contains hydrogel material, on visual acuity and long-term stability [83]. The results suggested that the sharp edges reduced PCO onset and improved visual acuity; however, a longer follow-up would help determine whether the hydrogel component may result in declining visual acuity due to gradual PCO onset [83]. Extended-depth-of-focus IOLs (EDOFIOL) provide ideal distance and immediate vision, owing to their extended far focus area [83,85]. When compared to multifocal IOLs, these provide greater visual acuity at far and immediate distances; however, EDOFIOLs have lower acuity at nearer distances versus multifocal IOLs [84].

### 8.2. Contrast Sensitivity

Contrast sensitivity is the ability to distinguish objects from their backgrounds [86]. Comparisons between different IOLs to determine which imparts the highest level of contrast sensitivity have been thoroughly explored [13]. Newer IOL designs attempt to improve contrast sensitivity by decreasing glare. Comparisons between aspheric (hydrophobic acrylic) and conventional silicone and acrylic lenses indicated that aspheric IOLs provided significantly better visual acuity and contrast sensitivity (measured by functional acuity contrast testing) [87].

As multifocal IOLs have become commercially available, their contrast sensitivity has been optimized [88]. It has been shown in Lee et al.’s two-month study involving 135 patients that adding hydrophilic components like HEMA to hydrophobic multifocal IOLs overcomes the drawbacks of hydrophobic IOLs [88]. No glistening or PCO were observed, though the study was only two months long [88]. Furthermore, this provided improved contrast sensitivity, reduced photic phenomenon, and increased presbyopia correction compared to non-HEMA-containing multifocal IOLs [88].

The findings from Fernandez et al.’s study involving contrast sensitivity defocus curves for a PCO evaluation of 63 patients supported the effect of PCO and photic phenomenon on contrast sensitivity. However, there were no such findings for the effect of glistenings on contrast sensitivity [87,88,89]. Chang and Kugelberg’s study involved 78 patients being followed up nine years postoperatively [90]. They found that though patients with hydrophilic IOLs had significantly fewer glistenings, the onset of glistenings was not correlated to an altered contrast sensitivity [90].

### 8.3. Long-Term Stability

IOL biocompatibility, within the context of resistance to long-term complications, seems more ideal with hydrophobic acrylic IOLs than with hydrophobic or PMMA IOLs. This can affect the IOLs’ long-term stability. Though IOLs are ideally expected to remain in patients’ eyes for the remainder of their life, certain factors, like PCO, can hinder their stability. Hydrophobic acrylic IOLs have shown greater resistance to phenomena like calcification and PCO [16,89]. Calcification and PCO degrade optical clarity over time [14,90]. Darcy et al. found that some manufacturers modify base acrylics to reduce calcification risk in hydrophilic acrylic IOLs [89]. Hydrophilic IOLs have been shown to be more likely to undergo ACO. Glistening formation is suggested to be more common in hydrophobic IOLs; however, glistening has not been found to correlate to decreased visual acuity [91].

Besides being biocompatible, long-term stability also involves a lack of dislocation from the implantation spot [92]. Mayer-Xanthaki et al. investigated the connection between IOL features and the IOLs’ likelihood of undergoing dislocation. Via reviewing medical records between 1996 and 2017, they investigated 68,199 eyes of 46,632 patients [92]. Most dislocations were in-the-bag (0.16%) instead of out-of-the-bag (0.05%) [92]. Hydrophilic IOLs had the highest in-the-bag dislocation, followed by quadripode IOLs and haptic angulation [92]. Three-piece IOLs and those with larger diameters were correlated to a lower dislocation risk, supposedly owing to their size being restrictive to their movement or dislocation [92]. Silicone IOLs were associated with a higher risk of out-of-the-bag dislocation, followed by hydrophilic IOLs [92]. This suggests that IOL materials may influence their chances of being dislocated postoperatively [92].

This was corroborated by Teshigawara et al.’s study involving 69 patients [93]. The authors attempted to expound the relationship between postoperative IOL shifts and refractive changes in order to predict how different IOL characteristics affect refraction postoperatively [93]. Knowing the predicted postoperative refraction (PPR) helps manufacturers remain mindful of changes in refraction when creating IOLs [93]. Three IOLs were considered: FEMTIS (hydrophilic acrylic plate with flanges for capsulorhexis fixation), FineVision (hydrophobic acrylic IOL), and AcrySoft IQ (hydrophobic acrylic IOL) [93]. Measurements were taken one day, one week, and one month post-surgery [93]. Besides IOL shift and refraction, the measurements included anterior chamber depth, axial length, and lens thickness, which were found to be useful predictors of IOL shift [93]. The results revealed that the IOL characteristics influenced refraction changes and postoperative IOL shifts [93]. FEMTIS had the lowest IOL shift and refraction [93]. This may be due to its feature of being fixed to the anterior capsule, and requiring capsulorhexis fixation [93]. Additionally, in the early postoperative stage, the capsulorhexis fixation technique for IOL implantation was found to render greater stability (less IOL shift and refraction changes) than in-the-bag techniques [93].

## 9. Analysis of Patient Outcomes

### 9.1. Surgical Outcomes, Complication Rates

IOL types are also another factor that surgical outcomes depend on [94]. Terveen at al. conducted a retrospective observational cohort study [94]. They investigated 133,896 eligible cataract surgery records’ postoperative outcomes, and compared between three different IOL types: standard monofocal, toric, and presbyopia-correcting IOLs [94]. They found that the overall incidence of Nd:YAG capsulotomy for PCO treatment was low overall, both within 6 months (4.7%) and within 12 months (8.7%) [94]. From those that underwent Nd:YAG capsulotomy, presbyopia-correcting IOLs had the highest proportion [38]. They also had the highest incidence of IOL exchange in six months or less when compared to other IOLs [94]. However, Bala et al. conducted a prospective, randomized, double-blind, parallel-group controlled clinical study which involved 282 patients; it compared presbyopia-correcting IOLs with aspheric monofocal IOLs across 19 sites in four countries [95]. Both groups had similar rates of clinically nonsignificant (subjective) PCO, clinically significant PCO, and YAG capsulotomy-requiring PCO [95]. This suggests that more research is needed to conclusively compare different IOLs for post-surgical outcomes [95].

### 9.2. Patient Satisfaction

Patient satisfaction has been noted to be influenced by the type of IOL used [96]. Hovanesian conducted a questionnaire-based study to evaluate patient satisfaction levels between 2 and 10 years post-cataract surgery [97]. Overall, 68 patients who received bilateral accommodating IOLs and 49 patients who received bilateral multifocal IOLs were questioned for the study [97]. The findings reported that most patients were satisfied with their choice of IOL when asked more than five years post-surgery [97]. Multifocal IOLs, nevertheless, presented with more glare and halos, as corroborated by other studies [83,84,97]. Niazi et al. compared patient satisfaction from using five different types of IOLs [96]. Their comparative study involved 164 patients who reported their satisfaction and quality of life using questionnaires [96]. Comparisons were made between premium (toric, extended depth of focus/multifocal) IOLs [96,98]. The findings, collected more than three years post-surgery, suggested that patients were more satisfied with multifocal IOLs, and that increasing numbers of patients are gravitating towards presbyopia-correcting IOLs [96].

Nevertheless, the fears of subsequent dysphotopsias and second operations have been found to discourage patients from seeking premium IOLs (premIOLs) [98]. Jameel at al. conducted a 12-question survey involving 360 patients using the Likert scale to determine patient satisfaction post-surgery, and their dependence on eyeglasses postoperatively [98]. In total, 85.8% of patients who underwent surgery were willing to continue wearing eyeglasses post-surgery [98]. This may be because 75.3% of respondents were simply unfamiliar with premIOLs, and 58.9% of them were reticent to opt for them due to potential risks [98]. This suggests that the method of using patient satisfaction to determine IOL effectiveness may be confounded by a lack of patient awareness about IOLs [98].

## 10. Challenges and Future Directions

### 10.1. Current Challenges in IOL Biomaterial Development

Figure 3 summarizes the recent innovations in IOL optimization, and the modifications made to conventional biomaterials. To overcome the drawbacks associated with traditional biomaterials, polymeric materials have been increasingly involved in IOL development to co-polymerize with the conventional biomaterials [30]. The key aims of current biomaterial development include achieving smaller incisions, imparting an adjustable ability to IOLs to improve visual quality, minimizing immune response to IOLs and fibrosis, and effectively dealing with complications like PCO [30]. This can be achieved via optimizing the use of flexible IOLs, ensuring patient-specific eye power, and improving biocompatibility, respectively. Finally, a major challenge is curbing the costs of IOL development to increase their accessibility [30].

### 10.2. Emerging Technologies

To optimize IOLs’ safety, biocompatibility, and efficacy, newer methods have emerged, including modified premIOLs, as well as innovative nanotechnology-dependent techniques. For example, Pedrotti et al. studied the visual acuity and astigmatism tolerance of 20 patients who underwent bilateral implantation of a premIOL with a continuous transitional focus element added to correct presbyopia [99]. The defocus curves produced from astigmatism measurements suggested a high tolerance for astigmatism [99].

#### 10.2.1. Nanotechnology

Babizhayev proposed using nanotechnology to coat IOLs with platinum film, which has antioxidant activity, to catalyze peroxide compounds’ reduction and decrease ROS levels in the aqueous humor [100]. Using nanotechnology was projected to be cost-effective and would ensure that thin coatings are compatible with the environment [100]. Also, it was suggested to mitigate cataract surgery-associated complications, like immune reactions and secondary cataracts [100].

The utility of nanotechnology, especially nanoparticles with organic coating (dendrimers, liposomes, nanoemulsions), in conferring antibacterial and anti-inflammatory properties to IOLs (Figure 4) has been extensively studied in the current literature [101]. The innovations in this realm are of an eclectic nature. They comprise many different techniques, including antibacterial nanopillar arrays on IOLs, photothermally activated anti-inflammatory and antibacterial nano-systems to prevent endophthalmitis, photodynamically activated supramolecular anti-keratitis nanoparticles that can be triggered by bacterial biofilms, and using gold nanoparticles to prevent bacterial or macrophagic adhesion to IOLs [102,103,104,105].

In situ forming gels are also a component of nanotechnology, wherein the injected gel responds to environmental changes in the conjunctival sac to form a viscoelastic gel [101]. This provides mechanical stability and biocompatibility [101].

#### 10.2.2. Future Innovations and Potential Impact on Cataract Surgery

The future of IOLs seems to be heading towards a greater focus on increased IOL biocompatibility, improved light transmission and visual acuity, increased durability, smart and dynamic biomaterials (shape-changing, photodynamic IOLs), and improved immunological responses to infection or inflammation (antibacterial-coated IOLs) [106]. The uniting feature of all these is innovating ideal delivery techniques for formulations to create the desired effect upon IOL implantation [101]. Cost-effectiveness and improved customizations are also being optimized as we consider the future of IOL innovations [106].

## 11. Conclusions

The development of innovative polymeric biomaterials has transformed the landscape of intraocular lens (IOL) design, addressing critical challenges in cataract surgery. This review has highlighted the essential requirements for IOL biomaterials, including biocompatibility, optical clarity, mechanical stability, and resistance to opacification, among others. Traditional materials like PMMA, silicone, and acrylics have laid the foundation, while cutting-edge polymers such as hydrogels, shape memory polymers, and light-adjustable lenses have introduced unprecedented versatility and performance. Additionally, computational modeling techniques are increasingly used to predict and optimize the mechanical, optical, and biocompatible properties of emerging polymeric IOLs, accelerating development and guiding material design before clinical testing.

Functionalization strategies, including surface modifications and drug-eluting designs, demonstrate significant potential to decrease the risk of complications like posterior capsule opacification (PCO), postoperative inflammation, and endophthalmitis. Emerging technologies, particularly nanotechnology and smart polymers, offer promising avenues for personalized, biocompatible, and high-performance IOLs. Clinical evaluations underscore the importance of optimizing visual outcomes, contrast sensitivity, and long-term stability to enhance patient satisfaction and quality of vision.

By integrating insights from material science, polymer engineering, and ophthalmology, and employing advanced computational modeling tools to predict and optimize IOL material properties, interdisciplinary work could potentially overcome current limitations and deliver increasingly sophisticated solutions. With continued collaboration between materials scientists and ophthalmic surgeons, the future holds immense promise for innovative, cost-effective, and patient-centric IOL technologies that redefine standards in cataract surgery.

## Figures and Tables

**Figure 1 jfb-15-00391-f001:**
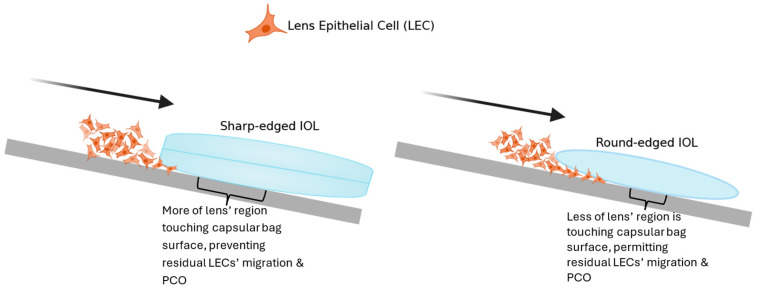
Due to the curvature of their lens, square (or sharp)-edged IOLs have a greater surface area that remains in contact with the capsular bag. This blocks the migration of LECs into the IOL’s posterior region, as seen on the left. Conversely, due to round-edged IOLs having a lower surface area in contact with the capsular bag, LECs can easily migrate to the IOLs’ posterior surface, resulting in PCO formation. This is shown on the right.

**Figure 2 jfb-15-00391-f002:**
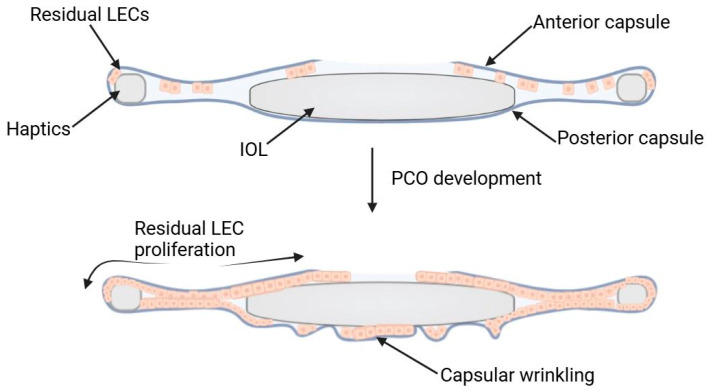
Residual LEC migration and proliferation followed by capsular wrinkling in the development of the fibrotic form of PCO.

**Figure 3 jfb-15-00391-f003:**
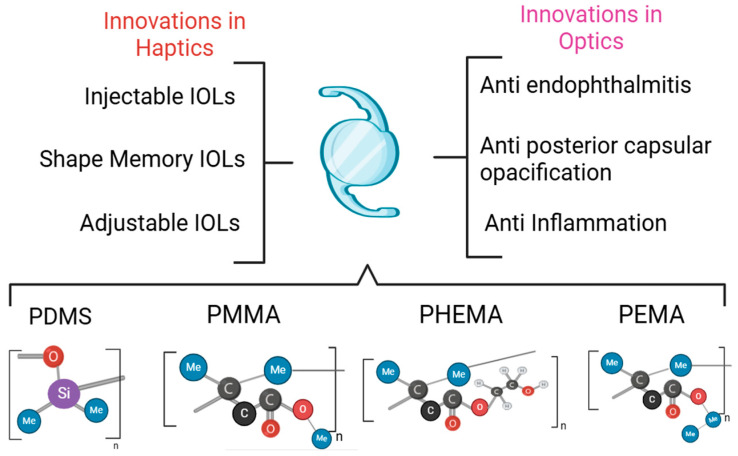
Summary of recent innovations in developing biomaterials for increased biocompatibility and IOL stability. Haptics and optics have both been extensively considered to produce improved overall patient outcomes. PDMS: polydimethylsiloxane; PMMA: polymethyl methacrylate; PHEMA: poly (2-hydroxyethyl methacrylate); PEMA: poly (ethyl methacrylate).

**Figure 4 jfb-15-00391-f004:**
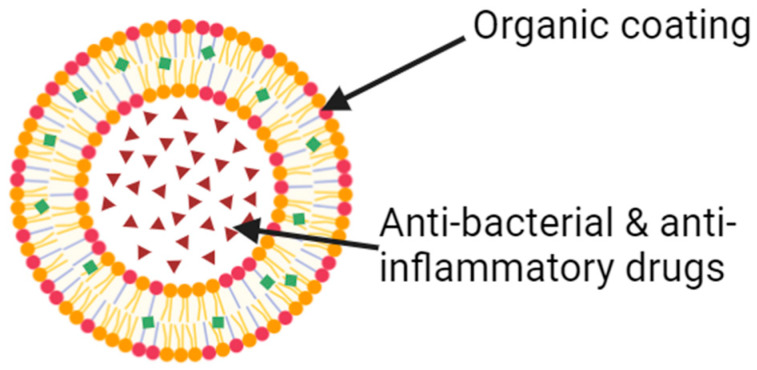
Diagram of a nanoparticle with an organic coating and the desired contents within the lipid membrane. This allows the nanoparticle to reach its destination before being uncoated. Research currently focuses on adding photodynamic components to these nanoparticles.

**Table 1 jfb-15-00391-t001:** Summary of cutting-edge polymer characteristics.

Materials	Biocompatibility	Specific Characteristics
Hydrogels	Great biocompatibility	High transparency Very flexible Can be 3D-printed
Shape memory IOLs	High uveal biocompatibility	Smaller surgical incisions High transparency Very minimal or no glistenings
Light-adjustable lenses (LALs)	Good biocompatibility	Stable refraction Good visual acuity No IOL-associated pathologies No significant opacification Adjustable post-insertion

**Table 2 jfb-15-00391-t002:** Summary of surface modifications to reduce PCO.

Surface Modification	PCO Prevention Mechanism	Example
Hydrophobic	Increased cell adhesion	RGD
Hydrophilic	Prevention of cell adhesion	PSBMA brush coating PEG
Photodynamic coating	ROS-induced apoptosis	α-CD-Ce6-PPEGMA brush
Photothermal modification	Thermal energy kills LECs	PDA
Drug loading	Antiproliferative drugs	DOX HA-Pac/CHI
	Antimetabolic drugs	MTX
	Immunosuppressant drugs	CsA-PLGA

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
