# Peer review of "Innovative Polymeric Biomaterials for Intraocular Lenses in Cataract Surgery"

_jfb, 2024, doi:10.3390/jfb15120391_

Round 1

Reviewer 1 Report

Comments and Suggestions for Authors

Good Day,

The fields of biomaterials is of great interest nowadays, especially the fields of the materials used for intraocular lenses in cataract surgery. Unfortunately, the cataract incidence increased in the last decades, and in the same time the age to develop the cataract decreased.

The Abstract is concise and describes the goal of the review. I appreciate that all the used abbreviations are explained atleast first time when they appear. Moreover, the similarity ratio / the percent obtained from the iThenticate report for the whole manuscript, is relatively low.

The used materials, and also their properties, are described in details. When the authors mention the hydrogels, there is no reference to hyaluronic acid or other biopolymers, which are also biocompatible materials, physiologically present on the human body. For instance, it is possible to use hyaluronic acid in an aqueous solution as a medical product, when the lenses are applied to the eye surface? Or, there is a need in general to use a gel or a solution when the lenses are introduced? And if so, what it should content?

The Nanotechnology chapter describes a very important and new direction in the medicine, for the future. Further on the manuscript, it is explained that the hydrophilic 2-methacryloyloxyethyl phosphorylcholine (MPC) can then be grafted onto the IOL. But this is the only mention of the MPC. It would be interesting for the readers to explain what exactly MPC means. It contains among other, phosphorus, a chemical element esential to the DNA. Nevertheless, the materials phosphorus containing could be also biocompatible. This should be pointed out. The authors could also mention more biocompatible compounds phosphorus containing, in addition to MPC. For instance there are hybrid materials with phosphorus, synthesized as gels, by the sol-gel method or by the surface grafting method, studied already for the treatment of different diseases and patologic conditions (for example osteosarcoma, as described in the work of Khaled et al. doi.org/10.3390/cimb46050290). Such compounds could be used as biocompatible materials in applications related to the eye health?

The manuscript is well documented and well written, sustained by the conclusions and by the references list, and it can be considered for publication after performing minor revision in agreement with the above-mentioned suggestions.

Best regards!

Author Response

Dear Reviewer,

We would like to sincerely thank you for your constructive and thoughtful comments, which have helped us improve the quality and clarity of our manuscript. Below, we address each of your comments in detail and explain the revisions we have made.

  1. Comment on Hyaluronic Acid and its Role in Ophthalmology
    “When the authors mention the hydrogels, there is no reference to hyaluronic acid or other biopolymers, which are also biocompatible materials, physiologically present in the human body. For instance, it is possible to use hyaluronic acid in an aqueous solution as a medical product, when the lenses are applied to the eye surface? Or, is there a need in general to use a gel or a solution when the lenses are introduced? And if so, what it should contain?”

Thank you for this insightful comment. We agree that hyaluronic acid (HA) is an important biopolymer with notable applications in ophthalmology. In our updated manuscript, we have clarified its role in Section 3.2 (“Cutting-edge Polymers: Hydrogels”), highlighting its biocompatibility and physiological presence in the vitreous humor. Additionally, we included a note that HA-based solutions are commonly used during cataract surgery (e.g., as viscoelastic substances) to protect ocular tissues when introducing IOLs.

However, we also clarified that hyaluronic acid itself is not currently used as a structural material for intraocular lenses but can serve as a surface modification for drug delivery, as mentioned in Section 4.1.

  1. Comment on MPC and Phosphorus-containing Biocompatible Materials
    “The hydrophilic 2-methacryloyloxyethyl phosphorylcholine (MPC) is mentioned, but further explanation of its significance and phosphorus-containing biocompatible materials would enhance understanding.”

We appreciate your suggestion to expand on the role of MPC. In our revised manuscript, we elaborated on the properties and significance of MPC in Section 4.1 (“Surface Modifications”). Specifically, we explained how the phosphate groups in MPC contribute to biocompatibility by mimicking the phospholipid bilayer of cell membranes, reducing protein adhesion and inflammation.

Furthermore, we incorporated your suggestion to mention other phosphorus-containing biocompatible compounds and their potential applications in ophthalmology. We referenced the work of Khaled et al. (doi.org/10.3390/cimb46050290) as an example of phosphorus-containing hybrid materials, such as sol-gel synthesized gels, and noted their possible relevance for ocular applications, particularly in drug delivery and surface coatings.

  1. General Comments
    We greatly appreciate your positive remarks on the abstract, level of detail, and quality of the manuscript. We also thank you for recognizing the low similarity ratio in the iThenticate report.

Your suggestion to expand on critical areas, such as hyaluronic acid and phosphorus-containing materials, has enhanced the comprehensiveness and value of our review.

Once again, thank you for your time, thoughtful comments, and appreciation of our work. We believe that the revisions we have made address all the concerns you raised, and we hope that the updated manuscript meets your expectations for publication.

Kind regards,

Reviewer 2 Report

Comments and Suggestions for Authors

Authors described polymeric biomaterials used for intraocular lenses in cataract surgery. It is an interesting work, however, some issues need to be further considered:

-          Please expand information on service life of the lenses, and the factors that influence it;

-          3.1 – PMMA – please correct the full name for “Poly(methyl methacrylate)”; “polymethylmethacrylate (poly EMA),” - ?;

-          Acrylic – please change to “Acrylic polymers” or “Acrylics”;

-          “Silicone is composed of dimethyl siloxane (DMS) monomers”  - it is not true, please change as silicones are a large family of various polymers;  in general, the polymer part contains some unusual terms (e.g. “Hydrophilic acrylic IOLs are composed of hydroxyethyl methacrylate (HEMA)”, “Hydrogels are emerging as cutting-edge polymers”, etc. ) and should be carefully checked by a polymer science expert;

-          SMPs and LALs in IOLs – some examples of polymers that are utilized?

-          3. 2 “Cutting-edge polymers” – please change for e.g. “IOL design”;

-           Kindly make it clear how IOL’s structure – properties relationships work in post-operative inflammation and infection mechanism;

-          Conclusions – what about modelling of these new materials and their properties?

Author Response

We sincerely thank the reviewer for their constructive feedback and thoughtful comments. We have carefully addressed all the points raised, and corresponding changes have been made in the revised manuscript. Below, we provide detailed responses to each specific comment.

  1. “Please expand information on service life of the lenses, and the factors that influence it.”
    • Thank you for this suggestion. We have clarified that the service life of intraocular lenses (IOLs) is typically for the remainder of the patient’s life unless complications such as posterior capsular opacification (PCO) arise. Additionally, we have further elaborated on long-term stability factors in Section 8.3 (Long-Term Stability).
  2. “3.1 – PMMA – please correct the full name for ‘Poly(methyl methacrylate)’; ‘polymethylmethacrylate (poly EMA),’ - ?”
    • We appreciate your attention to detail. The typo has been corrected, and the full name has been updated to Poly(methyl methacrylate).
    • The incorrect term "polymethylmethacrylate (poly EMA)" has also been corrected to poly ethylmethacrylate (poly EMA) under the acrylics section.
  3. “Acrylic – please change to ‘Acrylic polymers’ or ‘Acrylics’.”
    • We have updated the term to ‘Acrylics’ as per your recommendation.
  4. “Silicone is composed of dimethyl siloxane (DMS) monomers” - it is not true, please change as silicones are a large family of various polymers.
    • We appreciate your correction. The statement has been revised to:
      “Silicone is a polymer composed of dimethyl siloxane (DMS) monomers.”
      • Additionally, we have cited Vacalebre et al., 2023 (Reference 30) to support the revised description.
  5. “In general, the polymer part contains some unusual terms (e.g., ‘Hydrophilic acrylic IOLs are composed of hydroxyethyl methacrylate (HEMA)’, ‘Hydrogels are emerging as cutting-edge polymers’, etc.) and should be carefully checked by a polymer science expert.”
    • Thank you for pointing this out. We have carefully reviewed and refined the terminology in the polymer-related sections to ensure accuracy and clarity.
  6. “SMPs and LALs in IOLs – some examples of polymers that are utilized?”
    • We have added specific examples as follows:
      • In Section 3.2.2, we clarified that “LALs are composed of silicone and polymethylmethacrylate.”
      • In Section 3.2.1, we included an example of a solid SMP developed by Song et al., composed of ethylene glycol phenyl ether acrylate (EGPEA), ethylene glycol phenyl ether methacrylate (EGPEMA), and a long alkyl chain. Further details on its thermal properties, refractive index, and biocompatibility have been provided.
  7. “3.2 ‘Cutting-edge polymers’ – please change for e.g. ‘IOL design’.”
    • We have revised the heading to “Innovative Polymers for IOL Design” as suggested.
  8. “Kindly make it clear how IOL’s structure–properties relationships work in post-operative inflammation and infection mechanism.”
    • We have expanded Section 7 to clarify the relationship between IOL structure–properties and post-operative inflammation and infection mechanisms, providing a more comprehensive explanation.
  9. “Conclusions – what about modelling of these new materials and their properties?”
    • We have addressed this point by briefly discussing the importance and potential of computational modeling for predicting the behavior and properties of new IOL materials in the Conclusions section.

We are grateful for the reviewer’s valuable insights, which have helped us improve the manuscript significantly. We hope the revisions meet your expectations and look forward to your favorable consideration.

With our sincere thanks and appreciation,

Round 2

Reviewer 2 Report

Comments and Suggestions for Authors

Please replace

"Silicone is a polymer composed of dimethyl siloxane (DMS) monomers. It is hydrophobic, has a refractive index between 1.41 and 1.47, and requires an incision of at least 3.2 mm [30]."

by

"Silicones are a class of polymers mostly composed of covalently bonded silicon and oxygen atoms (siloxanes).  One of the widely used silicones is polydimethylsiloxane (PDMS) which is hydrophobic, has a refractive index between 1.41 and 1.47, and requires an incision of at least 3.2 mm [30]." 

Author Response

We thank the reviewer for their valuable suggestion to improve the clarity and accuracy of our manuscript. The text has been revised as requested. Specifically, we have replaced:

"Silicone is a polymer composed of dimethyl siloxane (DMS) monomers. It is hydrophobic, has a refractive index between 1.41 and 1.47, and requires an incision of at least 3.2 mm [30]."

with:

"Silicones are a class of polymers mostly composed of covalently bonded silicon and oxygen atoms (siloxanes). One of the widely used silicones is polydimethylsiloxane (PDMS) which is hydrophobic, has a refractive index between 1.41 and 1.47, and requires an incision of at least 3.2 mm [30]."

This revision has been incorporated into the updated manuscript.